# Cognitive Disengagement Syndrome and Child Sleep Problems in ADHD, Anxiety and Depression

**DOI:** 10.3390/healthcare11142022

**Published:** 2023-07-14

**Authors:** Almudena Cano-Crespo, Inmaculada Moreno-García, Mateu Servera, Manuel Morales-Ortiz

**Affiliations:** 1Department of Personality, Assessment and Psychological Treatment, Faculty of Psychology, University of Seville, 41018 Seville, Spain; imgarcia@us.es; 2Institut Universitari d’Investigació en Ciències de la Salut, Fundación Instituto de Investigación Sanitaria Islas Baleares, University of Illes Balears, 07122 Palma, Spain; mateus@uib.es; 3Department of Experimental Psychology, Faculty of Psychology, University of Seville, 41018 Seville, Spain; morales@us.es

**Keywords:** CDS, ADHD, sleep, children

## Abstract

The main objective of this study was to analyse the relationship and differential contribution of Cognitive Disengagement Syndrome (CDS) and sleep problems in children in different psychopathological measures. A total of 1133 participants (612 fathers/mothers and 521 children aged 7–13 years in Years 3–6 of Primary Education) completed the scales on CDS, sleep problems, depression (DEP), anxiety (ANX) and attention deficit and hyperactivity disorder (ADHD). Similar and significant correlations were identified in the measures of CDS and sleep problems between fathers and mothers, obtaining a high coefficient between the two parents. However, weak correlations were found between fathers/mothers and children. The scores of CDS and the sleep disturbance scale for children (SDSC) significantly predicted the internalising measures and ADHD, especially between SDSC and DEP. It was confirmed that sleep problems had a greater presence in the group of children with the highest ADHD scores, and especially in CDS and ADHD jointly. To conclude, the data indicate the importance of sleep problems in understanding CDS and its relationship with other psychopathological measures, especially ADHD, although to a greater extent with internalising symptoms, especially DEP.

## 1. Introduction

Sluggish Cognitive Tempo (SCT), currently known as Cognitive Disengagement Syndrome (CDS) [1,2,3], includes a set of symptoms characterised by excessive drowsiness, lethargy and mental confusion [1,4,5]. The reliability and convergent and discriminant validity of the items that evaluate CDS have been proved with other measures, especially the inattention (IN) of attention deficit and hyperactivity disorder (ADHD), which is strongly associated with [5,6,7]. Nevertheless, it is still necessary to determine the difference with other symptoms related to CDS [1]. Specifically, many cases show an overlapping between CDS and sleep problems, which can even be mistaken for central behaviours of CDS, such as mental confusion and tiredness [8].

A study conducted on a sample of university students found that ADHD and depression were related to a greater prevalence of suffering from sleep problems [9]. In a different study, it was found that IN and CDs were more strongly related to sleep problems than other symptoms, such as hyperactivity/impulsivity (HI). Specifically, CDS has shown a significant correlation, even controlling for the effect of inattention, with poorer sleep quality and a greater frequency of nightmares or night waking [10]. In the study by Langberg et al. [11], CDS predicted more diurnal sleep problems, even after controlling for the influence of other symptoms and disorders, such as ADHD, anxiety (ANX) and depression (DEP). Consequently, they found that CDS and sleep were different constructs in the factor analyses. Similarly, it was found that CDS and sleep problems reported by the parents (worse sleep quality, interrupted nighttime sleep and diurnal drowsiness) were also related to each other when controlling for the effect of the measures of ADHD [7]. The study conducted by Mayes et al. [12] demonstrated the association between CDS and sleep problems, as in the research carried out by O’Hare et al. [13], who detected that CDS was more strongly related to internalising symptoms and social isolation.

Studies conducted in samples of minors report that children in primary education were more prone to having sleep problems [14]. Another study that analysed CDS and sleep problems in minors [15] found a significant association between these two measures, although the convergence between the information from parents and children obtained moderate values. However, when controlling for the effects of the internalising symptoms and ADHD, only the relationship between CDS and the difficulties in waking remained significant. Likewise, in a sample of children diagnosed with inattentive ADHD, moderate correlation coefficients were obtained between CDS and sleep problems [16]. Nevertheless, these two studies showed two methodological issues: the use of sleep measures that had not been validated, and the inclusion of a sample that was exclusively composed of children with ADHD, excluding children with a normotypical profile [17]. In evaluations performed with adolescents with ADHD, it was observed that CDS, sleep problems, anxiety and depression were separate constructs [18]. With regard to the relationship between CDS and sleep problems in self-reported measures, a stronger correlation was observed for CDS with worse rest and diurnal drowsiness [19,20]. The consequences of a lack of sleep and a worse sleep quality [21,22] that predispose the child and adult population with ADHD and CDS were clearer in subjective measures than in objective measures [23]. However, the analysis of objective measures has contributed to some findings, even showing similar conclusions to those found with subjective measures and which would manifest this relationship between CDS and sleep problems in different psychopathological measures. Specifically, Fredrick et al. [24] identified a later nighttime circadian preference in children with CDS, as well as correlations between the lack of nighttime sleep and diurnal drowsiness, and between CDS and difficulties in regulating emotions [25].

Our study aims to analyse the relationship and differential contribution of Cognitive Disengagement Syndrome (CDS) and child sleep problems (evaluated by the parents and the child him/herself) in different psychopathological measures. Specifically, with the first objective, we aimed to analyse the convergence between the measures of the parents and those of the children themselves in CDS and sleep problems. These measures self-reported by the children, and their comparison with those reported by the parents, were included, following the work of Sáez et al. [26], expecting a moderate correspondence between the two informants. The second objective was to determine the differential or single predictive capacity of CDS and sleep problems in different psychopathological measures (DEP, ANX, IN and HI), after controlling for the sociodemographic variables of age and sex. We expected to obtain at least a stronger correlation between CDS, sleep problems and internalising symptoms, with sleep problems being established as an independent factor from the rest of the constructs [15]. Proceeding in a similar way to the work developed by Pagerols et al. [27], through the third objective, we compared the proportion of participants with sleep problems as a function of the clinical subgroups created from the scores obtained in the measures of CDS and ADHD in the parents, with the expectation of detecting more problems in the group of children with higher scores of ADHD.

## 2. Materials and Methods

### 2.1. Participants and Procedure

All the families from 9 schools in the province of Seville (Spain) whose children were studying in Years 3–6 of primary education were invited to participate in the study. The sample was constituted by 1133 participants, of whom 612 were parents (484 fathers and 586 mothers) and 521 were children (271 boys and 250 girls) aged 7–13 years (*M* = 9.99, *SD* = 1.43).

All the participating parents signed an informed consent form and completed the questionnaires. The families belonged to a medium socioeconomic level. The sociodemographic information was extracted from the variables “education level” and “professional, labour and marital status” of the questionnaire for parents.

The parents received the two evaluation protocols in an envelope and returned them two weeks after. We revised them one by one to verify who had completed at least one of the two scales, in order to consider them for the last phase of the evaluation of the children. The children completed the evaluation protocol in the classroom, with the help of two researchers, who guided them and solved doubts. Moreover, to minimise the number of losses, recovery sessions were held, which gathered the children who, in the day that was originally established in their centre, could not attend the first session and complete the questionnaire.

### 2.2. Instruments

Firstly, we present the instrument that was applied to the children and, secondly, the two scales for the parents:

The sleep self-report (SSR) [28] consists of 26 items, although we also included the Spanish factorisation with four subscales (routines to go to sleep, sleep-related anxiety, sleep quality and sleep refusal), constituting a total of 16 items [29]. Each item consists of 3 points (0 = almost never; 1 = sometimes; 2 = almost always). The greater the score, the greater the sleep problems.

The child and adolescent behaviour inventory (CABI) [30] for parents (https://tinyurl.com/CABI-Spanish) consists of 69 items. All items are evaluated in a 6-point scale from 0 (the child almost never presents the problematic behaviour) to 5 (the child almost always presents the problematic behaviour), except for the measures of difficulties of academic performance and social interaction, which range from 0 to 6. Cronbach’s alpha is 0.71 to 0.95 for all scales, demonstrating good reliability coefficients, as well as adequate structural and discriminant validity [31].

The sleep disturbance scale for children (SDSC) [32] for parents. This instrument is composed of 6 factors, with a total of 26 items: disorders of initiating and maintaining sleep (DIMS), sleep breathing disorders (SBD), disorders of arousal (DA), sleep–wake transition disorders (SWTD), disorders of excessive somnolence (DOES), and sleep hyperhidrosis (SHY). Each of the items is evaluated from 1 to 5 (1 = never; 2 = occasionally; 3 = sometimes; 4 = often; 5 = always). The greater the score, the greater the sleep problems.

### 2.3. Data Analysis

All statistical analyses were carried out with R v.4.2.2 software [33]. Firstly, a descriptive analysis of the variables was performed, and then, for the first objective, we obtained the correlations between the scores for the three sources of information of CDS and SDSC of the parents and SSR of the children. Proceeding with the second objective, regression models were carried out for each of the parents, which were validated through bootstrap procedures using the boot function of R [34]. A total of 1000 repetitions of each model were conducted, and the confidence intervals were constructed for each of the parameters of each model. The model was considered valid if each of the parameters was included in its corresponding confidence interval. The regression models included age and sex as control variables. Lastly, to reach the third objective and compare the proportion of the participants with sleep problems, as a function of the clinical groups created from the scores of the CDS and ADHD of the parents, the scores of the subscales and the total score of SDSC of both parents were dichotomised. Thus, the presence/absence of sleep problems was determined based on the cut-off points applied in the validation work conducted with a Spanish population [27], using the formula (*T-score* = 50 + [value − mean]/*SD* × 10), considering (*T-score* > 70) as the pathological threshold. The scores of the scales of ADHD (IN and HI) and CDS of the parents were also dichotomised, considering the 90th percentile that was originally established [35]. Four groups were differentiated according to the score below or above this 90th percentile in CDS and ADHD: Group 1 (control), which was constituted by participants whose scores were below the 90th percentile in CDS and ADHD; Group 2 (CDS), which consisted of participants with scores above the 90th percentile in CDS and below the 90th percentile in ADHD; Group 3 (ADHD), which was composed of participants with scores below the 90th percentile in CDS and above the 90th percentile in ADHD; and Group 4 (CDS + ADHD), which was constituted by participants with scores above the 90th percentile in CDS and ADHD. The groups were compared for the measures of fathers and mothers separately, applying Pearson’s Chi-squared hypothesis test. In some cases, when the requirements were not met, Fisher’s exact test was used, as recommended by Mehta and Patel [36]. Furthermore, the effect size was calculated through Cramer’s V. The data were expressed in contingency tables, and the interpretations derived from the adjusted residuals were correspondingly reflected in the results section.

## 3. Results

The correlations between the three information sources for the measures of CDS and SDSC of parents and the SSR of children were significant. The CDS and SDSC coefficients of parents were moderate. The association of the SSR of children with the SDSC and CDS of fathers/mothers was relatively weak. The convergence between fathers and mothers in the total measure of sleep (SDSC) was very high (*r* = 0.82, *p* < 0.001). Table 1 shows the correlations for fathers and mothers separately.

The regression models were validated, observing that the obtained parameters were included within the confidence intervals that corresponded to their value. Table 2 presents the standardised regression coefficients for CDS and SDSC on the scores of the other measures for each of the evaluators (father and mother), controlling for the effect of the variables of sex and age. The evaluations of both fathers and mothers of CDS and SDSC predicted the rest of the scores significantly (ANX, DEP, HI and IN). However, CDS predicted DEP to a greater extent (*r* = 0.22 in both parents) and IN (*r* = 0.69 in fathers and *r* = 0.66 in mothers), compared to the prediction of SDSC on DEP (*r* = 0.10 in fathers and *r* = 0.07 in mothers) and IN (*r* = 0.19 in fathers and *r* = 0.16 in mothers).

After comparing the proportion of children with sleep problems as a function of the groups created from the scores obtained in the measures of CDS and ADHD, significant differences were found between these groups in four of the subscales and in the total score that measured sleep problems in parents. According to the answers of the parents, the results show that the participants of Group 1 (control) presented low sleep problems in DIMS, SWTD, DOES and SDSC, whereas the groups that showed more difficulties were Group 3 (ADHD) in DIMS, Groups 3 and 4 (CDS + ADHD) in SWTD and SDSC, and Group 4 in DOES. Regarding the mothers, all scores of the subscales and the total scale of SDSC showed significant differences. In all cases, Group 1 presented low sleep problems, while Group 4 showed greater difficulties. Moreover, it was found that Group 3 (ADHD) presented more problems in DIMS, SBD, DA, DOES and SHY, and Group 2 (CDS) in SWTD and SDSC. Table 3 shows the different comparisons between the groups for each evaluator (fathers and mothers).

## 4. Discussion

This study aimed to analyse the relationship and contribution of CDS and sleep problems in different psychopathological measures, from a multi-informant perspective.

Therefore, firstly, we analysed the correlation between the measures of CDS (fathers and mothers) and the measures of sleep problems (fathers, mothers and children). Saéz et al. [26] found moderate correlations between the scores of parents and children, although the correlation coefficients obtained in our study were lower than expected. These conflicting results could be due to the fact that the included measures that evaluated sleep problems were different between evaluators, that is, SDSC in fathers and mothers and SSR in children. Furthermore, this would be in line with the findings of Orgilés et al. [29], who reported that the perception of the children toward sleep problems was different from that reported by the fathers/mothers. On the contrary, the correlation between fathers and mothers was very strong, which is in line with the results obtained in other studies [26,37,38,39].

With respect to the second objective, it was observed that the scores of the parents in CDS and SDSC predicted the internalising symptoms and ADHD significantly. This prediction of CDS, especially for internalising symptoms, is in line with those found in previous studies [1,10,40,41,42,43]. Our results are similar to those reported by Langberg et al. [44], since CDS predicted the measures of IN to a greater extent. Furthermore, in agreement with the findings of a study about the impact of sleep difficulties on the mental health of adolescents [45], sleep problems predicted a greater presence of not only ANX and DEP, but also HI and IN. However, in contrast to our results, these authors observed that sleep problems predicted more DEP, even one year after the end of the study.

In the third objective, our data show that the most common sleep problems in parents appeared in SWTD, DIMS and DOES, which is in line with the results obtained in the validation with a Spanish population [27], although these authors did not distinguish between fathers and mothers, thus it is not possible to know whether they only included one of the informants. Moreover, at the transcultural level, this greater prevalence of SWTD, DIMS and DOES was also found when the same instrument was applied to children aged 5–16 years in China [46] and Turkey [47]. Regarding the proportions of sleep problems found in the subscales SBD and SHY, they were similar to those found in the study of Pagerols et al. [27]; however, the percentages found in SHY in the present study were higher, which would be in agreement with the assertions of these authors, who stated that SHY is more prevalent among children in primary education. Lastly, the proportion of DA was greater in our study, which is in line with the results of previous studies [48,49,50], since younger children would present more nightmares and parasomnias. Furthermore, we found that the mothers reported a greater prevalence of sleep problems compared to the fathers, which is in agreement with the results obtained in studies that analyse the parental differences in the evaluation of child problems [51,52]. In addition, we observed that the groups of children with ADHD and, to a greater extent, the groups of children with ADHD and CDS jointly, presented more sleep problems, which is in line with the findings of previous studies [11,17,19,53].

The limitations of this study include the narrow age range of the children, since only the stage of primary education was included. Therefore, if the sample is extended to older ages, including adolescence, more information would be obtained about developmental changes. In this sense, it would be appropriate to add not only the sample from schools, but also other types of clinical samples with more extreme scores and different behavioural profiles. Furthermore, this study lacked the inclusion of teachers’ observations that could contribute to a better understanding of the problem by encompassing not only the family but also the school context. Future research lines could also incorporate other types of informants (e.g., teachers) and other measures (e.g., emotional regulation). Another recommendation for future research would be to conduct objective assessments of children’s sleep quality, as they were not included in this case and could provide valuable information about sleep problems in the child. On the other hand, it would also help to relate the information obtained from the three sources consulted and between the different types of measures (subjective and objective).

Among the main implications and contributions of this study, which could be derived from the findings of this research, would be, above all, the inclusion of a multi-informant perspective. According to the results obtained in this work, it is considered that this perspective would be the most appropriate option, because each of the evaluators could contribute enriching information on the measures included. Thus, the divergence found between parent and child assessments would imply that both, the ratings provided by the parents and self-reported measures by children themselves are relevant, since, on the one hand, parents would report the problems they perceive in their children, and, on the other hand, children would report the difficulties they perceive in themselves that parents have not detected or observed. Therefore, it is important not to overlook any case with underlying problems, especially when internalising symptoms are considered. Even if the correlation between fathers and mothers were very high, it would not be equivalent or completely coincident. In fact, depending on the measures, they are more or less concordant, which would mean that fathers and mothers perceive their children’s problems differently and could enrich the possible detection of difficulties in one direction or the other. Finally, this study compares different groups according to their ADHD and CDS scores, establishing different combinations. Consequently, it does not simply compare groups of children with more or less ADHD and CDS symptoms, but also includes what happens in the absence and presence of both. In this way, it is observed that there is a high number of problems not only in the ADHD group, but also when ADHD and CDS occur simultaneously. Regarding other clinical and practical implications of this research, the importance of understanding in more depth the relationship between CDS and sleep difficulties and other types of symptoms or disorders such as anxiety, depression and ADHD could be highlighted. Therefore, in the presence of any of these problems, clinicians could consider evaluating these associated symptoms. In addition, in the cases of children with ADHD and CDS, it would be important to detect if there are sleep problems, so that this could also be included in the design of possible treatment.

## 5. Conclusions

This study analysed the contribution of CDS and sleep problems in children from a multi-informant approach, observing a weak correlation between parents and children, and a strong correlation between fathers and mothers. The predictive capacity of the measures of CDS and SDCS on ANX, DEP and TDAH was demonstrated, with this relationship being clearer between SDSC and DEP. Lastly, a greater presence of sleep problems was corroborated in the group of children with the highest scores in ADHD and also in CDS and ADHD jointly.

## Figures and Tables

**Table 1 healthcare-11-02022-t001:** Correlations between the measures of CDS, SDCS and SSR of fathers, mothers and children.

	Fathers		Mothers	
	CDS	SDSC	CDS	SDSC
SDSC	0.36 ***	-	0.43 ***	-
SSR	0.10 *	0.18 ***	0.12 *	0.24 ***

*Note:* SDSC (sleep disturbance scale for children), SSR (sleep self-report), CDS (cognitive disengagement syndrome). * *p* < 0.05. *** *p* < 0.001.

**Table 2 healthcare-11-02022-t002:** Partial standardised regression coefficients of the measures of CDS and SDSC of fathers and mothers on other psychopathological and difficulty measures.

			Fathers					Mothers		
	*R^2^* adj	CDS	*BCa*	SDSC	*BCa*	*R^2^* adj	CDS	*BCa*	SDSC	*BCa*
ANX	0.27	0.15 ***	[0.10, 0.20]	0.12 ***	[0.07, 0.16]	0.24	0.14 ***	[0.10, 0.20]	0.10 ***	[0.06, 0.14]
DEP	0.37	0.22 ***	[0.17, 0.28]	0.10 ***	[0.05, 0.17]	0.37	0.22 ***	[0.18, 0.28]	0.07 ***	[0.03, 0.11]
HI	0.28	0.28 ***	[0.19, 0.37]	0.31 ***	[0.20, 0.41]	0.31	0.29 ***	[0.21, 0.37]	0.29 ***	[0.20, 0.38]
IN	0.59	0.69 ***	[0.62, 0.77]	0.19 ***	[0.11, 0.28]	0.58	0.66 ***	[0.57, 0.73 ]	0.16 ***	[0.07, 0.25]

*Note:* SDSC (sleep disturbance scale for children), CDS (cognitive disengagement syndrome), ANX (anxiety), DEP (depression), IN (inattention), HI (hyperactivity/impulsivity), *BCa* (Bias-corrected and accelerated Bootstrap for Confidence Interval at 95% of the parameter estimation), *R^2^* adj (*R^2^* adjusted). *** *p* < 0.001.

**Table 3 healthcare-11-02022-t003:** Proportion of participants with sleep problems as a function of the clinical subgroups created from the scores obtained in CDS and ADHD for fathers and mothers.

	**SDSC Fathers (%)**													
	DIMS		SBD		DA		SWTD		DOES		SHY		Total scale	
	No	Yes	No	Yes	No	Yes	No	Yes	No	Yes	No	Yes	No	Yes
1. Control	79.96	5.51	81.68	3.45	80.86	4.30	80.52	4.55	83.41	1.72	79.78	5.38	81.84	3.36
2. CDS	4.19	0.66	4.53	0.22	4.52	0.22	4.33	0.43	4.31	0.43	4.09	0.65	4.48	0.45
3. ADHD	3.08	1.32	4.09	0.43	3.87	0.65	3.46	1.08	3.88	0.65	3.87	0.65	3.36	1.12
4. CDS + ADHD	4.41	0.88	5.17	0.43	5.16	0.43	4.33	1.30	5.17	0.43	4.73	0.86	4.48	0.90
*p*	0.00		0.27		0.20		0.00		0.00		0.06		0.00	
*Cramer’s V*	0.19		0.07		0.09		0.21		0.17		0.11		0.22	
	**SDSC Mothers (%)**													
	DIMS		SBD		DA		SWTD		DOES		SHY		Total scale	
	No	Yes	No	Yes	No	Yes	No	Yes	No	Yes	No	Yes	No	Yes
1. Control	77.19	6.75	79.25	4.83	79.71	4.31	79.20	4.70	80.18	3.96	79.17	4.85	79	4.83
2. CDS	5.11	1.28	5.90	0.36	5.75	0.54	5.06	1.27	5.41	0.72	5.39	0.90	5.02	1.30
3. ADHD	3.47	1.28	3.76	0.89	4.13	0.54	3.80	0.90	3.60	1.08	3.77	0.90	3.90	0.93
4. CDS + ADHD	3.47	1.46	3.76	1.25	3.59	1.44	3.07	1.99	4.14	0.90	3.77	1.26	2.79	2.23
*p*	0.00		0.00		0.00		0.00		0.00		0.00		0.00	
*Cramer’s V*	0.21		0.19		0.21		0.29		0.20		0.19		0.32	

*Note:* SDSC (sleep disturbance scale for children), Group 1 (control), Group 2 (CDS), Group 3 (ADHD), Group 4 (CDS + ADHD), CDS (cognitive disengagement syndrome), ADHD (Attention Deficit and Hyperactivity Disorder), *p* (Fisher’s exact test), DIMS (disorders of initiating and maintaining sleep), SBD (sleep breathing disorders), DA (disorders of arousal), SWTD (sleep–wake transition disorders), DOES (disorders of excessive somnolence), SHY (sleep hyperhidrosis), No (without sleep problems), Yes (with sleep problems).

## Data Availability

The data presented in this study are available within the present article.

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
