# Peer review of "Cognitive Disengagement Syndrome and Child Sleep Problems in ADHD, Anxiety and Depression"

_healthcare, 2023, doi:10.3390/healthcare11142022_

Round 1

Reviewer 1 Report

This study aimed to examine the correlation and distinct contributions of cognitive disengagement syndrome (CDS) and sleep problems in children across various measures of psychopathology. A total of 1133 participants, including 612 fathers/mothers and 521 children aged 7-13 years in Year 3-6 of Primary Education, completed scales assessing CDS, sleep problems, depression (DEP), anxiety (ANX), and attention deficit and hyperactivity disorder (ADHD). The findings revealed significant and similar correlations in CDS and sleep problems between fathers and mothers. However, weak correlations were observed between fathers/mothers and children. The authors concluded that these results highlight the significance of sleep problems in comprehending CDS and its association with other psychopathological measures, particularly ADHD. The study was conducted well overall, and the manuscript was effectively written.

1.      The current cross-sectional design makes it challenging to establish causal relationships between sleep problems, depression, anxiety, and ADHD. Attention problems may be linked to sleep disturbances, while sleep deprivation can lead to daytime difficulties in concentration.

2.      This study lacked the inclusion of teachers' observations. Previous research has shown that ADHD symptoms evaluated by teachers tend to be more reliable and closer to assessing neuropsychological deficits.

3.      Objective assessments of children's sleep quality were not conducted. Furthermore, it would be valuable to discuss the clinical implications of this study and outline recommendations for future research. Should the ratings be solely provided by fathers, mothers, or both? Can the ratings from the children themselves be omitted?

Reviewer 2 Report

In the introduction, lines 59-60, you elucidated the insufficiencies of previous studies related to CDS, sleep problems and ADHD, “Nevertheless, these two studies showed two methodological… excluding children with a normotypical profile [17]” which are good to demonstrate the reasons why we need your study. But, the last sentence “Fredrick et al. [24] identified a later nighttime circadian preference… CDS and difficulties in regulating emotions [25].” seems a bit odd to be put there, because you did not state the purpose of putting the results of this study here. What’s the main purpose you would like to bring out? How do its results affect your study? How do it relate to your study?

Just a minor issue, you might state “ Our study aims to …” in lieu of “The general aim of the present work…”. Also, the paragraphs of the aim, first, second and third objectives can be combined together.

What is the model developed by Pagerols et al.? This model was suddenly mentioned here without explanation, which makes it hard to follow. Not to expect everyone to know what Pagerols et al’s model are and at least some explanation needs to be stated. Also, this model is related to one of your objectives and thus, it is necessary to clearly explain it.

In the presentation of results, “Regarding the first objective”, “ With regard to the second objective” and “In regard with the third objective” can be deleted.

This empirical study is a meaningful work with a large sample size to approve the relationship between CDS, and sleep problems in children with ADHD and their parents. Significant implications and contributions of this study can be mentioned in the discussion section.

Round 2

Reviewer 1 Report

No further question.

Author Response

Thank you very much for your review and positive assessment on the manuscript.

Reviewer 2 Report

Could you combine the following two paragraphs "Our study aims to analyse the relationship...psychopathological measures." and "Specifically, with the first objective...higher scores of ADHD."? 

"Our study aims to analyse the relationship...psychopathological measures." is one sentence only. Only one sentence in one paragraph is not quite appropriate. Also, these two paragraphs explained the purposes and aims of the study and thus it had better to combine into one paragraph.

Author Response

We thank you for your indication, indeed, following your suggestion, in line 75 paragraphs "Our study aims to analyse the relationship...psychopathological measures." and "Specifically, with the first objective...higher scores of ADHD” have been combined into one paragraph.